# *In Ovo* Feeding Techniques of Green Nanoparticles of Silver and Probiotics: Evaluation of Performance, Physiological, and Microbiological Responses of Hatched One-Day-Old Broiler Chicks

**DOI:** 10.3390/ani13233725

**Published:** 2023-12-01

**Authors:** Mervat M. N. Ahmed, Zienhom S. H. Ismail, Ibrahim Elwardany, Jayant Lohakare, Ahmed A. A. Abdel-Wareth

**Affiliations:** 1Department of Animal and Poultry Production, Faculty of Agriculture, South Valley University, Qena 83523, Egypt; mervat.elhadary@ksiu.edu.eg; 2Animal Production in Desert Area Program, Faculty of Desert Agriculture, King Salman International University, South Sinai 46612, Egypt; 3Department of Poultry Production, Faculty of Agriculture, Sohag University, Sohag 82524, Egypt; 4Department of Poultry Production, Faculty of Agriculture, Ain Shams University, Cairo 11566, Egypt; 5Poultry Center, Cooperative Agricultural Research Center, Prairie View A&M University, Prairie View, TX 77446, USA

**Keywords:** blood biochemistry, broilers, *in ovo*, microbial population, nanobiotechnology, nutrition

## Abstract

**Simple Summary:**

*In ovo* feeding technology using eco-friendly products can play a significant role in poultry production. The current study aimed to evaluate the early intervention of green nanoparticles of silver and probiotics to enhance the embryonic development and, subsequently, the health status of hatched chicks by decreasing pathogenic bacteria populations. This study showed that *in ovo* green nanoparticles of silver, probiotics, and their combination improved the productivity, health status, and microbial counts of hatched one-day-old broilers.

**Abstract:**

The aim of this study was to investigate the effects of the *in ovo* feeding of green nanoparticles of silver (Nano-Ag), probiotics, and their combination on hatchability, carcass criteria and internal organs, biochemical parameters, and cecal microbial populations in hatched one-day-old chicks. On day 18 of incubation, 250 live embryo eggs were weighed and randomly assigned to one of five treatment groups: a negative control group, a positive control group consisting of chicks injected with 0.2 mL physiological saline, a group consisting of chicks injected with 0.2 mL Nano-Ag, a group consisting of chicks injected with 0.2 mL probiotics (*Bifidobacterium* spp.), and a group consisting of chicks injected with 0.2 mL combination of Nano-Ag and probiotics (1:1). The results showed that the *in ovo* injection of Nano-Ag or probiotics, alone or in combination, had no effect on hatchability, live body weight, or internal organs but improved (*p* < 0.05) chick carcass yield compared to the control groups. Furthermore, *in ovo* feeding decreased (*p* < 0.05) serum levels of cholesterol, triglycerides, urea, creatinine, alanine aminotransferase, and aspartate aminotransferase, as well as cecal *E. coli*, but increased *Bifidobacterium* spp. when compared to the control groups. Based on these findings, *in ovo* injections of green Nano-Ag and probiotics, either alone or in combination, have the potential to improve chick health and balance the microbial populations in hatched one-day-old chicks.

## 1. Introduction

Since the incubation period impacts embryonic growth, hatchability, the chick immune system, and post-hatch performance, *in ovo* technology can play an important role in poultry production as the primary source of pre-hatch nourishment for chicken embryos [1]. Improved poultry performance, lower antibiotic use, less condemnation at slaughter, and higher product quality are all positively correlated with better first-day-old chicken quality [2]. Pre-hatch nutrition for chicken embryos is just as important as post-hatch nutrition since incubation time affects embryonic development, hatchability, the chick immune system, and post-hatch performance [3]. Due to the importance of one-day-old chick health status and based on the above, research was conducted to investigate the effect of *in ovo* techniques or the early delivery of bioactive substances on poultry performance [4]; however, there is a dearth of information on day-old chick health status due to such nutritional interventions. This is based on the straightforward idea of providing the chick embryo with bioactive supplements to create lifetime characteristics in the bird, such as enhanced performance, immunity, and a healthy gut microbiome [5]. Antibiotics as growth promoters (AGP) in feed are banned in Europe, the United States, and a few other nations, including China, which outlawed this practice in 2020 [6]. The concepts of “nanobiotechnology”, and “nanobiology” all related to the integration of nanotechnology and biology [7]. Green nanoparticles of silver (Nano-Ag) are known to be produced by a variety of microorganisms, and the majority of these particles are spherical [8]. According to a few studies, feeding livestock and poultry nanoparticles may enhance their performance by boosting immune and digestive health [9]. One of the most promising probiotics, *Bifidobacterium* spp., has lately been employed as a substitute growth promoter in chickens mainly due to its involvement in stabilizing the gut microbiota and controlling the spread of infections, which is based on the competitive exclusion mechanism [4]. 

Beneficial bacteria can be introduced into the embryonic gut prior to hatching to help chicks manage stress during hatching, grow, use feed more effectively, and improve nutrient digestibility and absorption, decreasing mortality and lessening the burden of pathogenic diseases [3]. There is an opportunity to create viable AGP substitutes for chicken due to a recently developed area of *in ovo* technology that involves delivering bioactive substances directly to the developing embryo [10]. *In ovo* vaccination has been shown to elicit an early immune response in newborn chicks in contrast to post-hatch immunization [11]. Apart from fostering favorable immune reactions in avian species, *in ovo* technology has the potential to mitigate perinatal nutritional deficits in birds. These deficits are often caused by the transition from embryonic yolk nutrition to exogenous feeding [12]. Furthermore, this technology provides a chance to promote the development of the embryonic gastrointestinal tract [GIT], gut-associated lymphoid tissue, and the colonization of the embryonic gut with a healthy microbiota [5]. Due to their proven antibacterial capability, nanoparticles of noble metals (primarily silver) are now being used as disinfectants in animal production [13]. *In ovo* technology is essentially a biotechnological intervention that is typically adopted to ensure early immunological programming in birds [10]. Probiotics and nanoparticles of silver have been investigated as in-feed antibiotic substitutes; however, there have been no investigations on their potential symbiotic effects. Therefore, this study aimed to investigate the effect of using the *in ovo* feeding of green Nano-Ag and probiotics and their combination on performance, carcass criteria, blood biochemical parameters, and cecal microbial population in one-day-old broiler chicks. 

## 2. Materials and Methods

### 2.1. Macroalgae Selection, Collection, and Extract Preparation for Nano Silver Synthesis 

The macroalga Jania rubens was hand-picked from the Red Sea in Hurghada, Egypt. Epiphytes, extraneous debris, and necrotic algae were eliminated from healthy algal samples. Samples were thoroughly cleaned with sea water, then sterile distilled water, air-dried, chopped into small pieces, and crushed in a tissue grinder (IKA A 10, Schönwalde-Glien, Germany) until a fine powder shape was achieved. Dried seaweed powder [1 g] was combined with 100 mL of distilled water and heated to 100 °C before being filtered using Rotilabo^®^ Tyb 601P filter paper (Carl Roth, Karlsruhe, Germany), with the retained particles measuring 10 µm.

### 2.2. Green Synthesis of Silver Nanoparticles 

Silver nanoparticles were essentially produced using the same method as that described in [14]. A volume of 50 mL of 1 mM AgNO_3_ solution was mixed with 50 mL of Jania rubens algal extract under continuous stirring at 45 °C. Within 2 h, the solution’s color changed from brownish yellow to light purple, implying the development of Nano-Ag. The resulting solution was stirred for 4 h more to complete the reaction. The silver-nanoparticles and J. rubens mixture was separated from the leftover seaweed by collecting the pellets after 10 min of centrifugation at 6000 rpm/min. The pellets were suspended in double-distilled water once more, and the pH was adjusted by adding 0.1 mL of phosphate buffer to the entire volume to achieve physiological pH. This solution contained 0.17 mg Ag per mL. The particle’s structural morphology was examined using a transmission electron microscope (Figure 1).

### 2.3. Probiotic Strain Preparation

Bifidobacterium strains were obtained from Cairo Microbiological Resource Center (MIRCEN), Faculty of Agriculture, Ain Shams University. These bacterial strains’ standard inoculums were prepared via the inoculation of conical flasks (100 mL in volume) containing 50 mL of MRS broth medium with a loop of tested strains. The inoculated flasks were incubated at 37 °C for 72 h. One mL of this culture contained about 2 × 10^8^ cfu/mL of *Bifidobacterium angulotum*, *Bifidobacterium animalis*, and *Bifidobacterium bifidum*. 

### 2.4. Experimental Design and Incubation Conditions

Fertilized eggs (Ross 308) were obtained from a commercial hatchery from a 46-week-old breeding flock. The fertility of the eggs was confirmed by examining the growth of the embryos at 24 and 48 h of the embryos’ lifespan. After 10 days, optical examination (candling) was performed to exclude dead embryos and unfertilized eggs. Eggs bearing live embryos were randomly reallocated to experimental groups (50 eggs in each; five repetitions) with approximately comparable average egg weights (62.3 ± 21 g). After 408 h (17 days) of embryogenesis and based on previous reports [15,16], a total of 250 eggs with a live fetus were randomly assigned to one of five experimental groups. The first group acted as a control (no injection), while the second served as the vehicle control (injected with sterilized Saline Solution (0.2 mL per egg)), and the 3rd, 4th, and the 5th groups were injected with Nano-Ag at 0.2 mL per egg, probiotics (*Bifidobacterium angulotum* + *Bifidobacterium animalis* + *Bifidobacterium bifidum*) at 0.2 mL per egg, and a combination of Nano-Agat 0.2 mL and probiotics at 0.2 mL per egg, respectively. The dosages used for the probiotics were determined based on previous studies [4,5]; similarly, we also consulted the literature to select the dose of Nano-Ag [9]. After disinfecting the eggs with 70% ethanol, a hole was punched into the shell at the wider end of the egg (air-cell chamber) using a 22-gauge needle. Following the injection of the eggs, the injection holes were sealed with a sticky solution, and the eggs were placed in an incubator, ensuring that each treatment was equally represented in each region of the incubator. At 17.5 d of incubation, all eggs except the negative control were injected at the broader end of the egg into the amniotic fluid according to the procedure described by De Oliveira et al. [1]. The fertilized eggs were kept in a cool location at a constant temperature of 10 °C. The temperature and humidity inside the incubator were adjusted to keep the temperature at 37.5 °C and the relative humidity at 60 percent from day 1 to day 17 and then to 70 percent on day 18. The eggs were turned automatically in the incubator once every 6 h for the first 18 days, and turning stopped from day 19 until chicks hatched. According to the recommendations of the scientific research ethics council, this study was conducted during the pre-hatch growing period of broiler chickens.

### 2.5. Productive Performance

At hatch, the number of live hatched chicks and non-hatched chicks was counted to determine hatchability (%). All hatched chicks were weighed, and six chicks from each treatment were randomly selected for sampling.

The following formula was used to determine the hatchability (%):$$Hatchability\;[\%]=\frac{\left(Number\;of\;hatched\;fertilized\;injected\;eggs\right)}{Number\;of\;fertilized\;injected\;eggs}\times 100$$

### 2.6. Carcass Criteria

After weighing, the chicks were slaughtered per treatment group; the weights of the carcasses and weights of their internal organs (liver, gizzards, heart, proventriculus, intestine, and cecum) of the slaughtered chicks from each group were recorded and expressed as percent of relative live body weights. The cecum was stored at −80 °C in freezer for microbiology testing. 

### 2.7. Blood Sampling and Laboratory Analyses

Blood was obtained from the jugular vein during slaughtering and transferred in Vacutainer ^®^ tubes at the end of the experiment for serum collection. The blood was centrifuged for 30 min (1500× *g*) at room temperature. The serum was collected in tubes and kept at −20 °C until it was analyzed. Colorimetric analyses of liver enzymes such as aspartate aminotransferase (AST) and alanine transaminase (ALT), total cholesterol, triglycerides, as well as kidney function tests regarding urea and creatinine and total serum protein concentrations, were performed using commercial kits (Bio-diagnostic, Cairo, Egypt).

### 2.8. Microbial Enumeration

The cecum of the slaughtered birds was used to count the colony forming units (CFU) of *Escherichia coli* (*E. coli*) and *Bifidobacterium* spp. in the cecum. The digesta samples were separated and thoroughly mixed in separate sterile plastic containers. Ten grams of homogenized sample were mixed with ten-fold serial dilutions of physiological NaCl-Trypton in a stomacher bag and forcefully shaken for three minutes. MRSA agar for *Bifidobacterium* spp. was used as the plate media, and *E. coli* bacteria were counted by inoculating a 10-fold serial dilution of rinses onto *E. coli* petrifilms (3 M Corporation, St. Paul, MN, USA). In accordance with the manufacturer’s instructions, sterile saline (0.85%) was used for dilution. Typical *E. coli* and *Bifidobacteria* colonies were enumerated after 24 h of incubation at 35 °C.

### 2.9. Statistical Analysis

Our statistical analysis was performed using a completely randomized design and the general linear models (GLM) procedure of SAS 9.2 [17]. Data were analyzed by carrying out a one-way ANOVA. Duncan’s multiple range tests were used to compare means. Graphs were made in GraphPad Prism software, version 9 (GraphPad Software, La Jolla, CA, USA). All data were evaluated for normal distribution (W > 0.05) using the Shapiro–Wilks test. The eggs and chicks (replicate) were the experimental unit for the performance parameter evaluation. For carcass measurements, blood biochemistry, and caecal microflora, each broiler was the experimental unit. Values are expressed as mean and standard error of the mean (SEM). Significance was declared at *p* < 0.05, and a tendency toward significance was declared at 0.05 < *p* < 0.10. *p*-values less than 0.001 are expressed as “<0.001” rather than the actual value. 

## 3. Results 

### 3.1. Productive Performance

The effects of applying Nano-Ag, probiotics, and their combination *in ovo* on hatchability (%) and the hatched weight of broiler chicks are summarized in Figure 2. The results showed that the *in ovo* application of Nano-Ag, probiotics, and their combination did not negatively affect (*p* ≥ 0.05) hatched live body weight or hatchability (%).

### 3.2. Carcass Criteria and Internal Organ Weight 

The effects of the *in ovo* application of Nano-Ag, probiotics, and their combination on the carcass and internal organ weights of one-day-old, hatched chicks are presented in Table 1. The results indicated that the *in ovo* application of Nano-Ag, probiotics, and their combination did not affect (*p* ≥ 0.05) internal organ weight (liver, heart, gizzards, intestine, cecum, and proventriculus), while carcass weight was increased (*p* ≤ 0.003) in the probiotics and combination groups compared to the control groups but not different from the Nano-Ag group.

### 3.3. Blood Biochemistry

There was decrease (*p* < 0.001) in ALT and AST enzymes in one-day-old, hatched chicks in all treatments compared with the control group, as presented in Figure 3. Figure 4 shows the results regarding the effect of treatments on kidney functions; the results refer to a significant decrease (*p* < 0.001) in urea and creatinine levels, and the decrease in creatinine levels noticed in all the three treatment groups compared to the controls. The results for the one-day-old chicks (presented in Figure 5) showed a decrease (*p* < 0.001) in total cholesterol and triglyceride levels in the Nano-Ag, probiotics, and their combination treatments compared to the controls, but the lowest values were found in the combination treatment. As shown in Figure 6, among the one-day-old chicks, total protein was not affected (*p* ≥ 0.05) by the treatments involving early feeding via the *in ovo* technique.

### 3.4. Microbial Counts

Figure 7 shows the effects of the treatments on the total counts of *E. coli* and *Bifidobacterium* spp. The treatments decreased (*p* ≤ 0.001) *E. coli* counts; on the other hand, the total counts of *Bifidobacteria* increased compared to the controls (*p* ≤ 0.001) due to treatment with Nano-Ag, probiotics, and their combination. The highest decrease in *E. coli* counts and the highest increase in *Bifidobacteria* were recorded in the combination group compared to the other groups.

## 4. Discussion

Based on the significance of one-day-old chick health, various researchers have conducted studies to ascertain the impact of using an *in ovo* method or the early administration of bioactive chemicals on chicken performance. However, there is a dearth of information on the application of *in ovo* nanobiotechnology in poultry production. Our study examined the effects of Nano-Ag, probiotics, and their combination on day-old broiler performance, carcass criteria, serum biochemical parameters, and cecal microbial population in an attempt to find a novel way to raise birds’ immunity and general health status. In this study, neither hatchability (%) nor live chick weights were affected (*p* ≥ 0.05) by the *in ovo* application of Nano-Ag, probiotics, and their combination compared to the control groups, as described in Figure 2. Similarly, the *in ovo* injection of probiotics did not affect the hatchability in the studies by the authors of [1,18,19,20]. Similar to our study, one-day-old chick weight and hatchability (%) were not impacted by *in ovo* Nano-Ag compared to the control group in [21]. Directly injecting bioactive compounds into the growing embryo in order to produce superior lifelong effects while considering the dynamic physiology of the chicken embryo lies at the heart of *in ovo* technology [12]. The improvement found in our study could be due to the antimicrobial activity of Nano-Ag affecting harmful intestinal bacteria; improving gut health and the absorption of nutrients and stimulating digestive enzyme activity are other proposed indications of the Nano-Ag growth stimulatory effect. Similarly, in another study, the *in ovo* injection of probiotics (specifically injecting into the yolk sac) on day 17 of embryogenesis caused a significant increase in live body weight compared to that of the control group [4]. The pre-hatch colonization of the embryonic gut with beneficial bacteria can therefore assist the chicks in better managing stress during hatching, improving their growth, enhancing feed utilization, improving nutrient digestibility and absorption, reducing mortality, and reducing the burden of pathogenic diseases [4].

The effects of the *in ovo* application of Nano-Ag, probiotics, and their combination on the carcass and internal organ weight values of day-old hatched chicks are presented in Table 1. These results indicated that the application of *in ovo* injections of Nano-Ag, probiotics, and their combination did not affect internal organ weight (liver, heart, gizzard, intestine, cecum and proventriculus), while carcass weight increased in the three treatment group compared to the control groups in our study. These results agree with those of Duan et al. [22], who stated that after the *in ovo* administration of probiotics, there were no differences in immune organ weight after the *in ovo* injection, while the probiotics group’s villus height of the duodenum, jejunum, and ileum layers increased significantly in comparison to the non-injected and saline groups. Similarly, the supplementation of probiotics in a broiler diet from day one of life, resulted in statistically different weight values for their livers, hearts, and abdominal fat [23]. Additionally, Nano-Ag injection had no discernible impact on the liver, heart, kidney weight in rabbits, and there were no differences in dressing percentage or carcass weight [24,25]. On the other hand, compared to the control treatment, the *in ovo* injection of *Bifidobacteria* dramatically improved intestinal morphology, and this improvement could be related to higher digestive enzyme activity, as well as an increased small intestine segment absorptive surface area [4]. In general, there is a paucity of reports that mention the effects of the injection process in general on the carcass or internal organs of broilers, and this may be attributed to the lack of importance of these standards at one day of age. However, in our study, it was necessary to investigate the effects of the treatments and the injection process on the weight of the internal organs to ensure that any early symptoms, such as hepatitis or any other manifestations of deformity, could be recorded. Once there is liver damage or a myocardial infarction brought on by toxins or viruses, the activity of liver enzymes increases, and they are discharged from hepatocytes into the blood [26]. Figure 2 shows a significant decrease in ALT and AST enzymes in all treatment groups compared with the control group. This decrease may indicate that the liver status was good enough in all hatched chicks. These results are consistent with those of Dosoky et al. [27], who reported that Nano-Ag hydrocolloids had an impact on protein catabolism, as evidenced by decreased liver enzyme activity (ALT and AST). In alignment with our findings, researchers from our group, Lohakare and Abdel-Wareth [28], observed that Nano-Ag decreased plasma AST and ALT levels in broilers when compared to their control group. A slight decrease in the activity of the ALT and AST enzymes, responsible for directing amino acids onto catabolic pathways and lowering the plasma concentration of the primary byproducts of protein metabolism (creatinine and urea), may indicate disturbed protein catabolism in chickens fed Nano-Ag [26]. This contradiction in results may be due to the method(s) applied, sampling age, the nanoparticles synthesis method, or other nutritional factors. 

Similarly, exposure to Nano-Ag resulted in a decrease in aspartate aminotransferase (AST) activity in the blood plasma of chickens in [24]. It is well known that excessive concentrations of ALT in the serum indicate the development of the organ dysfunction and disease progression, with liver disease being the major reason for the increase in ALT [29]. Broilers supplemented with probiotics showed enhanced alanine aminotransferase (ALT), aspartate aminotransferase (AST), and alkaline phosphatase levels compared to the control group in [30]. On the other hand, plasma AST and ALT enzyme levels were unaffected by probiotic addition in [31]. Enhanced liver function was identified in the sera of probiotic treatment-broilers, as were reduced ALT and AST concentrations, which indicated the significant efficacy of bacilli in the protection of treated birds from hepatocellular damage compared to the controls. A slight decrease in the activity of the enzymes alanine aminotransferase (ALT) and aspartate aminotransferase (AST), responsible for directing amino acids onto catabolic pathways and lowering the plasma concentration of the primary byproducts of protein metabolism (creatinine and urea), may indicate disturbed protein catabolism in chickens receiving Nano-Ag [26]. 

The current study’s results show a significant decrease in urea and creatinine levels, and decreases in creatinine levels were noticed in the three treatment groups compared to the controls. The administration of probiotics resulted in linear decreases in urea and creatinine [32]. This may be a good indicator that Nano-Ag did not affect negatively kidney tissue or function or the organs of the urinary system. Nano-Ag hydrocolloids had an impact on protein catabolism, as evidenced by decreased liver enzyme activity (ALT and AST) and decreased levels of urea and creatinine [27]. Nanomaterials with a size range of 10–100 nm stay in the bloodstream for a long time before being carried to the organs, and nanoparticles can retain and collect in the body rather than be excreted, and this may affect the kidney’s physiological functions and, consequently, urea and creatinine levels [33]. On the contrary, Nano-Ag had no effect on urea or creatinine levels in broilers when compared to the control groups [28]. Adding multi-Bacillus strains lowered blood uric acid levels compared to the controls, but no significant variations in uric acid levels were identified between the groups. Uric acid is a nitrogenous excretory product of protein metabolism in birds, and its serum levels are measured as part of a renal function test [30]. These variations could be attributable to the application method, dose, or type of nanoparticles. This finding indicated that bacilli probiotics lowered renal pressure by lowering serum non-protein nitrogen to levels at the same level as uric acid [34]. Our results showed a significant decrease in total cholesterol and triglyceride levels in the treatment groups, and the highest decrease was found in the combination treatment compared to the others. As shown by a decrease in LDH activity, oxidative stress caused by Nano-Ag often affects mitochondrial function [35]. Cholesterol is included in developing cell membranes and bile formation and is a precursor of vitamin D and many hormones. Serum cholesterol levels in the probiotics group were lower than the control group in [30]. Also, Nano-Ag had a detrimental impact on blood lipid profiles, increasing cholesterol, low-density lipoproteins, and triglycerides in [24], contrary to our results. Microorganisms like *Lactobacillus* and *Bifidobacterium* work together to produce short-chain fatty acids, which lowers the levels of blood cholesterol, plasma cholesterol, and total lipids [31]. Kumar et al. [32] revealed that birds that were treated with probiotics had lower mean triglyceride levels and low-density lipoproteins than those in the control group. By reducing the blood cholesterol levels in hens, probiotic dietary supplements had a beneficial impact on the health of the host animal in [36,37,38]. In the current study, serum total protein was not affected by Nano-Ag, probiotics, and their combination in early feeding via the *in ovo* technique. Most nanoparticles damage proteins, enzymes, and nucleic acids in the mitochondria after they enter cells, causing lipid peroxidation [26]. Nano-Ag hydrocolloids had an impact on protein catabolism, as evidenced by decreased liver enzyme activity (ALT and AST) and decreased levels of urea and creatinine [24]. Further studies are needed on the effects of Nano-Ag on serum total proteins to elucidate the exact mechanism of the effects observed.

Similar to our results, heat-inactivated probiotics had no significant effects on total protein, albumin, or lipid contents in the blood of the animals in [39]. Similarly, probiotic supplementation in a broiler diet had little to no effect on blood plasma parameters such as globulins, total protein, and albumin in [40]. The antibacterial compounds released by bacillus that prevent the growth of infections and improve the utilization of food proteins were thought to be the cause of the higher blood albumin levels in birds receiving bacilli supplements in [34]. On the contrary, broilers given probiotics showed an increase in the levels of total protein and albumin when compared to the controls in [30]. Also, broilers fed diets supplemented with biological additives showed increased blood total plasma protein, globulin, hemoglobin, and albumin levels in [41]. In another study, birds supplemented with *B. subtilis* exhibited increases in the levels of albumin and total proteins in the blood, contradicting our results [29]. 

In the present study, decreased (*p* ≤ 0.001) total *E. coli* counts were noticed, but the total counts of *Bifidobacteria* were increased in the treated groups compared to the control groups. The highest decrease in *E. coli* bacteria and the highest increase in *Bifidobacteria* were recorded in the combination group compared to the other groups. The *in ovo* injection process had a significant impact on the microbial environment in chickens’ guts; this was accompanied by a significant increase in *Bifidobacterium* spp. and lactic acid bacteria counts, as well as a decrease in the population of total coliform bacteria in the ileal digesta [4]. Additionally, early gut microbiota stimulation with probiotics, prebiotics, and their combination can improve the health and productivity of freshly hatched chicks [42]. Additionally, *in ovo* inoculation with *Bacillus* spp. significantly reduced the total number of Gram-negative bacteria at the day of hatching and day 7 of age compared to the control group in [43]. Probiotics were found to boost the intestinal population of lactic acid bacteria on day 3 after hatching while decreasing the population of *E. coli* in [1]. Via the *in ovo* distribution of vegetative *Bacillus* spp. strains (included in Norum), the severity of virulent *E. coli* cross-infection in broiler chickens can be reduced [41]. Similarly, the viability of the *E. faecium* (M74) strain for injected probiotics via amniotic fluid injection on day 18 was shown by the authors of [44]. The *in ovo* administration of various probiotic species did not significantly affect the incidence of avian pathogenic *E. coli* in broiler chickens in [45]. The *in ovo* inoculation of probiotics was found to be effective at lowering pathogenic bacteria colonization from day 1 to day 7 in [46]. Compared to the control group, the *in ovo* injection of probiotics had no effect on an *E. coli* population in one-day-old chicks but reduced it on day three of age in [18]. Exposure to pathogenic bacteria during the perinatal period affects the growth capacity and immune responses of newly hatched chicks by impairing the development of immune organs, the GIT, and skeletal muscles. Moreover, Nano-Ag has biocidal capabilities that have been shown to reduce the overall quantity of bacteria as well as the number of dangerous *E. coli*, *Salmonella*, and *Streptococcus* species [13]. Nano-Ag showed antibacterial activity against *E. coli* by causing the cell walls to degrade at minimum inhibitory concentrations of 50 and 100 ppm, respectively [47]. 

## 5. Conclusions

In conclusion, the *in ovo* feeding of green Nano-Ag particles and probiotics, either alone or in combination, has the potential to improve chick health and balance the microbial populations in day-old hatched chicks. Moreover, the implementation of nanobiotechnologies such as *in ovo* feeding strategies may have favorable effects on performance, carcass, blood, and microbiological parameters in one-day-old chickens. However, additional studies on growing chickens for extended growth periods are still required.

## Figures and Tables

**Figure 1 animals-13-03725-f001:**
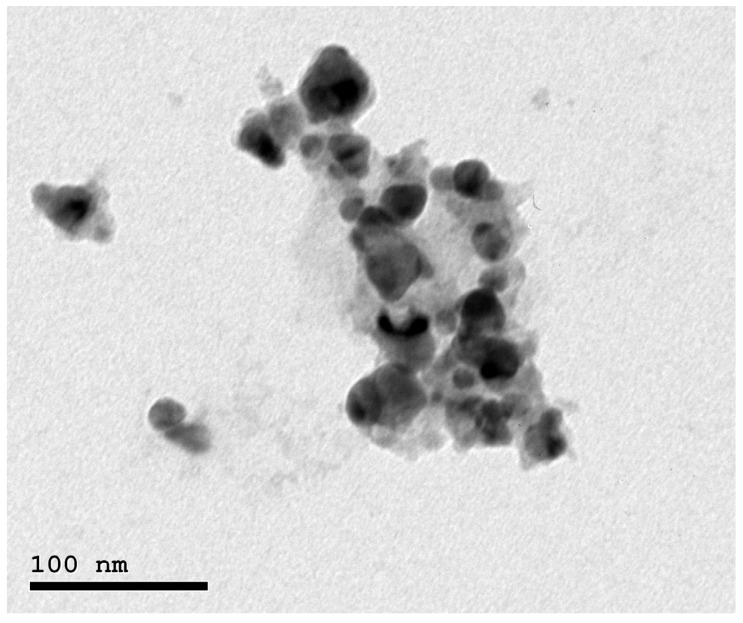
Transmission electron micrographs of green silver nanoparticles (Nano-Ag); 1–100 nm, and TEM Mag = 8000×.

**Figure 2 animals-13-03725-f002:**
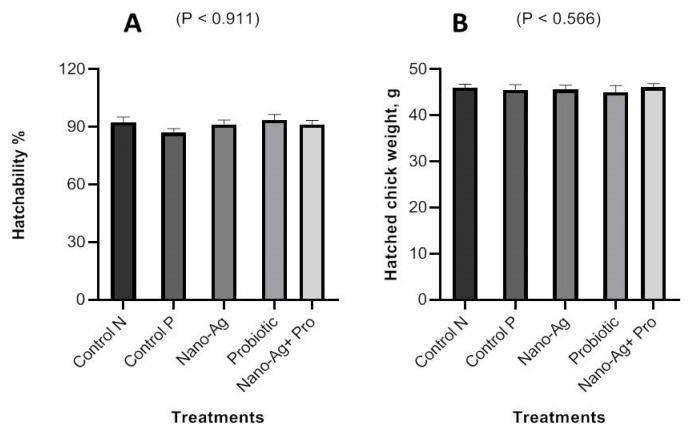
Effect of Nano-Ag, probiotics, and their combination on hatchability (%) (**A**) and hatched chick live body weight (**B**) in one-day-old hatched chicks. The bars on each column in the figure show the standard error of means (n = 50).

**Figure 3 animals-13-03725-f003:**
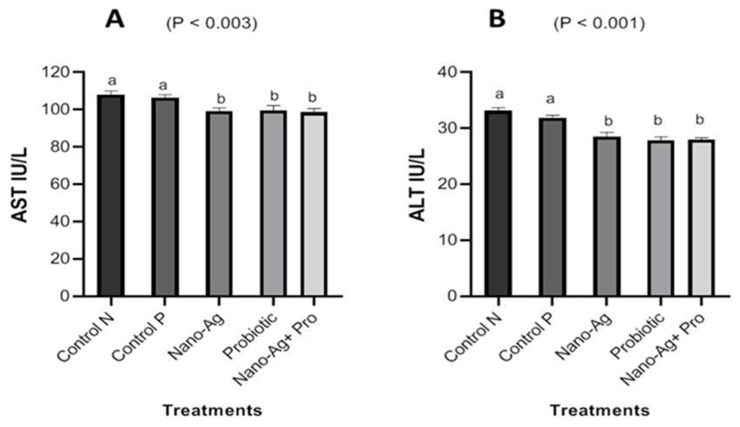
Effect of Nano-Ag, probiotics, and their combination on aspartate aminotransferase (AST) (**A**) and alanine transaminase (ALT) (**B**) levels in one-day-old hatched chicks. The birds were the experimental unit (n = 6 per treatment), and the bars on each column in the figure refer to the standard error of means. ^a,b^ The columns in figures with different superscripts are different (*p* ˂ 0.05).

**Figure 4 animals-13-03725-f004:**
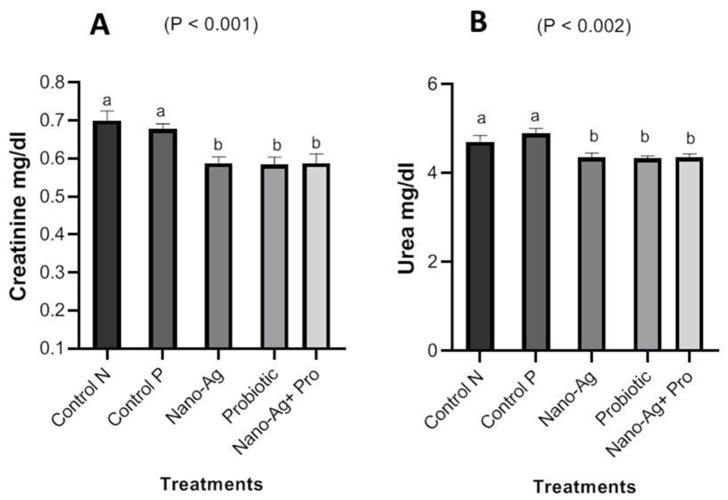
Effect of Nano-Ag, probiotics, and their combination on creatinine (**A**) and urea (**B**) levels in one-day-old hatched chicks. The birds were the experimental unit (n = 6 per treatment), and the bars on each column in the figure refer to the standard error of means. ^a,b^ The columns in figures with different superscripts are different (*p* ˂ 0.05).

**Figure 5 animals-13-03725-f005:**
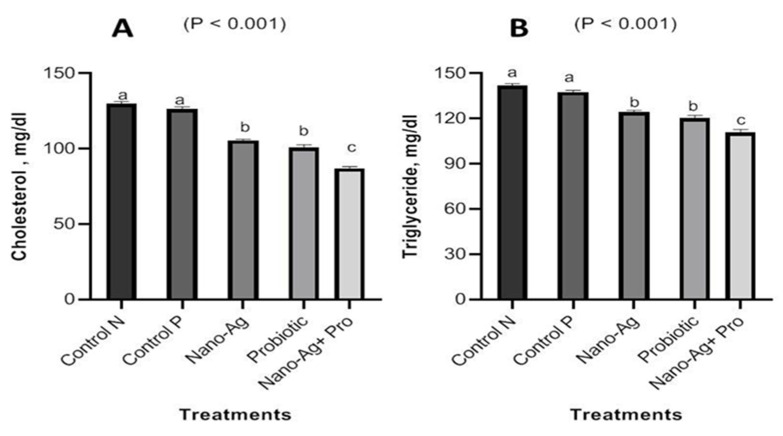
Effect of Nano-Ag, probiotics, and their combination on cholesterol (**A**) and triglycerides (**B**) levels in one-day-old hatched chicks. The birds were the experimental unit (n = 6 per treatment), and the bars on each column in the figure refer to the standard error of means. ^a–c^ The columns in figures with different superscripts are different (*p* ˂ 0.05).

**Figure 6 animals-13-03725-f006:**
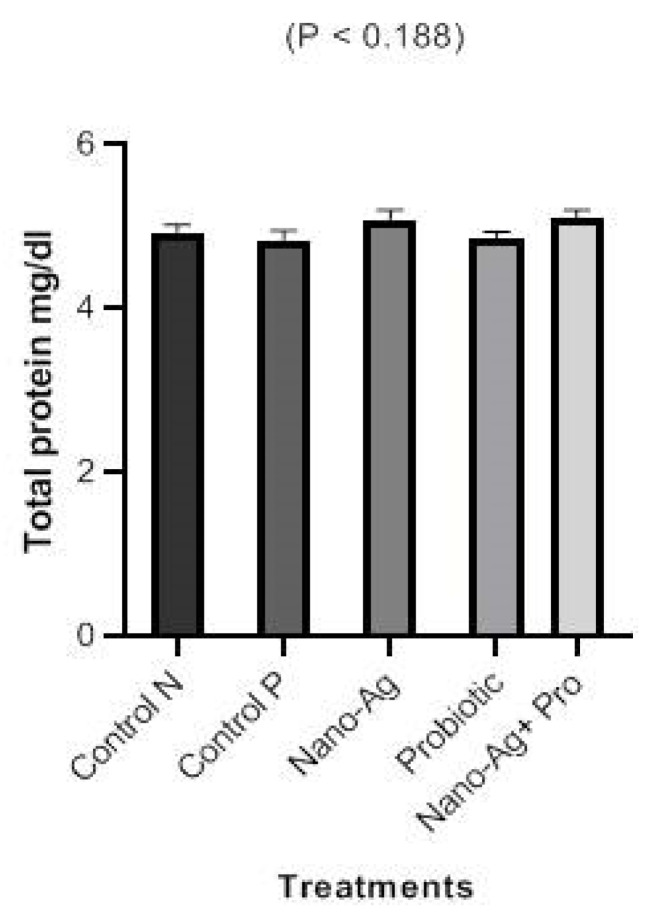
Effect of Nano-Ag, probiotics, and their combination on total protein levels in one-day-old, hatched chicks. The birds were the experimental unit (n = 6 per treatment), and the bars on each column in the figure refer to the standard error of means.

**Figure 7 animals-13-03725-f007:**
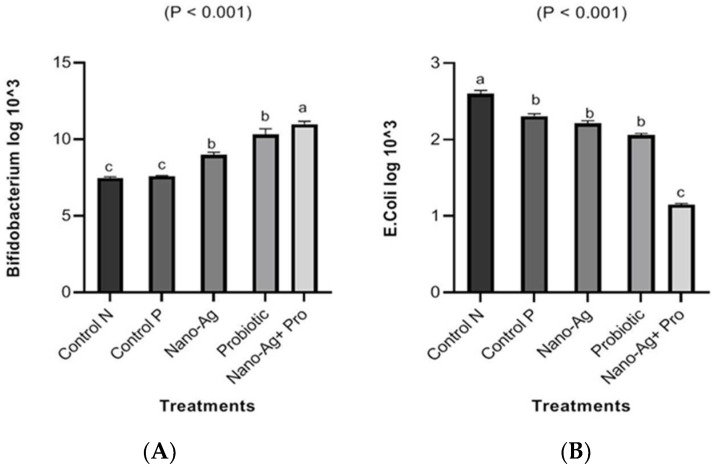
Effect of Nano-Ag, probiotics, and their combination on total counts of *Bifidobacterium* (**A**) *E. coli* (**B**) (CFU/g cecal content) in one-day-old hatched chicks. The birds were the experimental unit (n = 6 per treatment), and the bars on each column in the figure refer to the standard error of means. ^a–c^ The columns in figures with different superscripts are different (*p* ˂ 0.05).

**Table 1 animals-13-03725-t001:** Effect of Nano-Ag, probiotics, and their combination on the carcass and internal organ weight of hatched chicks (expressed as percentage of live weight).

Items	Treatments					SEM	*p*-Value
	Control-N	Control-P	Nano-Ag	Probiotics	Combination		
Carcass%	32.091 ^b^	33.713 ^b^	38.122 ^ab^	47.196 ^a^	41.558 ^a^	1.577	0.003
Liver%	2.842	2.753	2.970	2.847	2.866	0.042	0.669
Heart%	0.643	0.708	0.669	0.692	0.674	0.009	0.236
Gizzard%	4.883	5.209	5.127	5.109	5.144	0.103	0.916
Intestine%	4.226	4.397	4.154	4.568	4.532	0.105	0.721
Cecum%	1.424	1.427	1.580	1.583	1.579	0.045	0.647
Proventriculus%	0.996	1.049	0.935	1.137	1.037	0.029	0.273

^a,b^ Means within the same row carrying different superscripts are significantly different (*p* < 0.05) (n = 6 birds per treatment). Control N = control negative [non-injected eggs]. Control P = control positive [eggs injected with saline solution]. Nano-Ag [*in ovo*] = [injection with silver Nanoparticles at 0.2 mL/egg]. Probiotics [*in ovo*] = [injection with probiotics at 0.2 mL/egg]. Combination: Probiotics + Nano-Ag [*in ovo*] = injection mixture of Nano-Ag and probiotics at 0.2 mL/egg. SEM, standard error of means.

## Data Availability

The datasets used and/or analyzed during the current study can be made available from the corresponding author upon reasonable request (A.A.A.A.-W.).

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
