# Peer review of "In Ovo Feeding Techniques of Green Nanoparticles of Silver and Probiotics: Evaluation of Performance, Physiological, and Microbiological Responses of Hatched One-Day-Old Broiler Chicks"

_animals, 2023, doi:10.3390/ani13233725_

Round 1
Reviewer 1 Report (Previous Reviewer 2)
Comments and Suggestions for Authors
No comments
Author Response
|
Dear Respective Reviewer, thank you very much for your positive opinion on our manuscript and previous inputs which has improved the final form of our manuscript. |
Reviewer 2 Report (Previous Reviewer 4)
Comments and Suggestions for Authors
In each table and figure, the statistical unit should be noted.
Comments on the Quality of English LanguageMinor editing of English language required
Author Response
"Please see the attachment"

Reviewer 3 Report (Previous Reviewer 1)
Comments and Suggestions for Authors
This revised version of the manuscript is in better shape that the originally submitted. Authors did address the concerns that I had on my original revision and enriched the manuscript with more detailed descriptions, especially the statistical analysis, the origin of the probiotics and the enhancement of the conclusion. Nonetheless, when reading again the manuscript a couple of extra concerns that I did not pose on my original revision came to my mind:
Major concerns
1.- What was the procedure of the authors to define that 250 eggs (50 eggs per treatment) and 6 chicks were representative enough to conduct their analyses? In other words, how the authors determined that they had enough statistical power? Did the authors performed some sort of sample size calculation to define the number of eggs/animals used on the study? If so, authors should include that information to provide more certainty about their rationale. If not, I believe authors will need to explain why they chose to work with these numbers (maybe funding limitations could explain it) and clearly state that they results should be interpreted with caution due to the limited sample size used.
2.- The other important concern is related to the definition of the doses for the treatments used. More than just explaining the volume injected to the eggs, I believe that it is more important to explain what dose of the Nano-Ag and Probiotics was chosen and why. I consider this important because it is highly probable that the results observed on the study depend on the actual dose of the treatments. Without this piece of information, it may be difficult to extrapolate the results or even replicate the results. I respectfully recommend to the authors to address this point so it could be more understandable why you conducted the study in the way you did.
Minor edits
L35: In ovo should be in italics
L237: In ovo should be in italics
Comments on the Quality of English LanguageOther than minor edits, I consider the English of the manuscript understandable.
Author Response
"Please see the attachment"

Reviewer 4 Report (New Reviewer)
Comments and Suggestions for Authors
Important contribution describing effects of in ovo provision of silver nanoparticles and probiotics on performance and metabolic parameters. The work is entirely dealing with measurements carried out with one-day-old chickens. In this context, it is recommended to modify the title by adding “one-day-old chickens” or hatched one-day-old chickens”. Also, in the abstract.
The results are clearly presented, however, the discussion ought to be modified. Several results are compared with experiments performed with chickens (and rabbits) during a longer growth period, for example references: 9, 21, 22, 23, 24, 25, 26 45, 46. Whether present results can be directly compared with performance and metabolic parameters of older chickens is doubtful.
The main conclusion (L461-463) is partly justified, however, the following (L463-466) is not discussed, but it is a general presumption. The conclusions should outline the effects on performance, carcass, blood and microbiological parameters in one-day-old chickens. Further, indicate that additional studies with growing chickens during extended growth period are still required.
minor
Silver nanoparticles are abbreviated Ag-NPs, Nano-Ag. Use the same abbreviation at first occurrence (L27, L106), and then consequently in the whole text.
L92: delete ross.
L200: “performed separately for each data” the statement is not clear.
Figures: explain error signs above columns, SEM or SD.
Figure 6: “levels in one day old, hatched chicks” use the same wording as in other figures “levels in hatched chicks”.
Author Response
"Please see the attachment"

Round 2
Reviewer 3 Report (Previous Reviewer 1)
Comments and Suggestions for Authors
This version of the manuscript has addressed the major concerns I had before, authors provided references to justify their sample size and also backed up the dosages used on their treatments based on previously published literature. I consider that the article has improved enough to warrant publication. Perhaps my only two extra requests to the authors are:
1.- Add a sentence in section 2.4 in where you clearly show that your sample size is comparable to other studies. It could be something like: “After 408 h (17 days) of embryogenesis and based on previous reports (Elwan et al., 2022; Kollmansperger et al., 2023), a total of 250 eggs with a live fetus were randomly assigned to one of five experimental groups”. In this way, you will be providing certainty to the readers regarding the sample size used.
2.- Modify the sentence in section 2.4 regarding the election of the doses to avoid being repetitive, I suggest: “Dosages used for the probiotics were determined based on previous studies [4-5], similarly, we also relied on literature to select the dose of Nano-Ag [9].”
Comments on the Quality of English LanguageOther than minor edits, I consider the English of the manuscript understandable.
Author Response
"Please see the attachment."

Reviewer 4 Report (New Reviewer)
Comments and Suggestions for Authors
Thank you for modifying the paper. All my comments have been considered.
Author Response
|
Comments 1: Thank you for modifying the paper. All my comments have been considered. |
|
Response 1: Dear Respective Reviewer, thank you very much for your positive opinion on our manuscript and previous inputs which has improved the final form of our manuscript.
|
This manuscript is a resubmission of an earlier submission. The following is a list of the peer review reports and author responses from that submission.
Round 1
Reviewer 1 Report
Comments and Suggestions for Authors
I consider that the authors should add a paragraph explaining in detail the statistical analysis for all the variables studied. I am pleased with the explanation of the experimental design; however, no details were provided regarding the statistical model employed and the specific tests performed to compare the treatment means. Furthermore, authors should indicate which statistical software was implemented to conduct the analysis. This section could be inserted at the end of materials and methods as section 2.8.
There was a nice description related to the isolation and generation of Nano-Ag, but where do the authors describe the obtention of the probiotics (Bifidobacterium angulotum + Bifidobacterium animalis + Bifidobacterium bifidum)? Is there a commercial product available containing these microorganisms? Moreover, why these particular microorganisms and strains were chosen to use a probiotic on this study? I consider there is a lack of clarity of why these bacteria were selected as treatment and how they were actually obtained. Please expand on this and provide more clarity on the reasons behind using this probiotic as treatment.
I consider that some sentences should be added on the conclusion of the paper regarding the relevance and/or feasibility of using this nanobiotechnology approach in the poultry industry. It seems that the results of the study were considerably sound and important, so it could be reasonable to expand on this section in terms of explaining the potential implications and cost-benefits of adopting the in ovo feeding strategy more widely.
Minor edits
Authors should be careful when writing scientific names or Latin-based expressions. Sometimes authors do use italics with the expression “in ovo”, but in many other times they don’t. The same happen with the names of microorganisms (Escherichia coli, Bifidobacterium spp) or plants (Jania rubens). Please be consistent and use the appropriate way (italics) every time that you need to use a Latin-based expression or a scientific organism name. Go thru all the manuscript and fix all the inconsistencies.
L115: if you are trying to refer to an average weight why do you present a range? (57 to 77 g). I recommend mentioning the actual average ± standard deviation.
L117: the word candling do not require to be capitalized.
L128: this line is not clear, please rephrase. Are you trying to say that the Lab temperature and humidity were adjusted before the injection to try to be as close as the incubator as possible?
L132: When you say, “eggs remained outside the incubator for about the same time as much as possible”, how much time are you referring too? Maybe it would be better to somehow mention how long it took in average to perform the application of treatments.
L152: No need to repeat that the 6 checks were slaughtered, you already made that clear on line 150.
L158: were you trying to say Vacutainer ®?
L167-169: please rephrase this sentence as currently is not clear.
L180: since you are talking in plural, you should start with “The effects…”
L188: I believe you want to use a different word to start the paragraph, table 1 is not showing “the application”, is showing “the effects of the in ovo application”
Table 1: why the units of the variables are expressed as percentages? Weren’t they measured directly in grams? Please clarify
Figure 2: I suggest writing the full name of the enzymes on the footnote so the reader can easily understand the y-axes names.
L255-257: These two sentences are giving the same information. Please combine into one.
Please be consistent with the way of how you present significances, sometimes you don’t leave spaces (e.g., P<0.05) and sometimes you do (e.g., P < 0.05). Review the journal author guidelines and use the appropriate method.
Comments on the Quality of English LanguageIn general, the quality of the English writing of the paper is good. Other than minor edits, I believe is well written.
Reviewer 2 Report
Comments and Suggestions for Authors
- Lack of Structural Characteristics: The paper does not provide information about the structural characteristics of silver nanoparticles. This information is crucial for understanding the properties and potential effects of the nanoparticles used in the study.
- Limited Number of Eggs Set: The number of eggs set in the hatchery is noted to be too few. This raises concerns about the representativeness of the sample and the statistical power of the study. A larger sample size would enhance the validity and generalizability of the findings.
- Insufficient Detail on Egg Fertility Confirmation: The paper lacks detail on how the eggs were confirmed to be fertile. This is a critical aspect, as fertile eggs are essential for conducting experiments related to embryonic development.
- Provide Specific Dosing Information: Include detailed information about the concentration or count of nanoparticles and probiotics used in the study. This should specify the exact amount administered to the subjects, which is essential for accurately replicating the experiment.
-
Addressing this point will improve the clarity and reproducibility of the study.
-
Expand Conclusion: Take the opportunity to elaborate on the significance of the results and their broader implications. Discuss how the findings contribute to the existing body of knowledge in the field and suggest potential avenues for future research.
Reviewer 3 Report
Comments and Suggestions for Authors
Manuscript is rather long although the data are not substantially affected by the treatments. Discussion needs to be shortened for clarity.
Statistical analysis is lacking how data produced and analyzed.
50 eggs per treatment are not considered sufficient. It is not clear how hatchability was calculated with 50 eggs. It should have been replicated.
Post-hatch performance should be conducted. It is more practical to monitor post-hatch performance whether in-ovo nutrition indeed affects post-hatch performance such as body weight gain, feed conversion ratio, or gut development.
Sampling 6 hatched chicks per treatment are not considered sufficient.
Comments on the Quality of English LanguageIt is recommended to be English-edited.
Reviewer 4 Report
Comments and Suggestions for Authors
1. The simple summary is suggested to rewrite to briefly describe aims, major findings, and conclusions; too much introductory descriptions.
2. In Materials and Methods, line 95-101, what is the size of seaweed powder after filtered through Rotilabo® Tyb 601P filter paper?
3. The authors need to justify why the formulated nanoparticle called Nanoparticles of Silver (Nano-Ag) instead of Nanoparticles of Silver nitrate? How the authors ensured all of the silver of nitrate were release after interaction with macroalgae ?
4. How much dose instead of volume should be expressed in the ovo injection? For example, 0.2 mL of Nano-Ag at ? mg/mL, probiotics, at ? mg/mL, and combination of Nano-Ag and probiotics (1:1). Also, since the formulated Nano-Ag is derived by reaction of AgNO3 with organic algae, the authors need to calculated the % of siliver of nanoparticles based on wt/wt.
5. How many rounds of artificial hatch were made? If only 1 round how the authors calcaulated SD or SE?
6. How the authors excludes the concern by the publics due to the toxicity of residual silver if the slaughter carcass, when the green Nano-Ag is used as a feed additive?
7. Where are the chick immune response results?
8. In Figure 6, what is the unit based? CFU/g cecal content?
Comments on the Quality of English LanguageMinor editing of English language required